# Global Trends and Scientific Impact of Topical Probiotics in Dermatological Treatment and Skincare

**DOI:** 10.3390/microorganisms12102010

**Published:** 2024-10-03

**Authors:** Ademilton Costa Alves, Sergio Murilo da Silva Braga Martins, José Victor Trindade Belo, Mauro Victor Castro Lemos, Carlos Emanuel de Matos Chaves Lima, Carlos Drielson da Silva, Adrielle Zagmignan, Luís Cláudio Nascimento da Silva

**Affiliations:** Laboratório de Patogenicidade Microbiana, Universidade CEUMA, São Luis 65075-120, MA, Brazil; professorademiltonalves@gmail.com (A.C.A.); sergiomurillo21@gmail.com (S.M.d.S.B.M.J.); josevictor.belo1943@gmail.com (J.V.T.B.); maurocastro832@gmail.com (M.V.C.L.); carlos_emanuelll@hotmail.com (C.E.d.M.C.L.); drielsonn.sousa@gmail.com (C.D.d.S.); adrielle004602@ceuma.com.br (A.Z.)

**Keywords:** topical probiotics, skin microbiome, wound healing, inflammatory skin diseases, skin care

## Abstract

The skin plays a crucial role in maintaining homeostasis and protecting against external aggressors. Recent research has highlighted the potential of probiotics and postbiotics in dermatological treatments and skincare. These beneficial microorganisms interact with the skin microbiota, modulate the immune response, and enhance the skin barrier, offering a promising therapeutic avenue for various skin conditions, such as acne, dermatitis, eczema, and psoriasis. This bibliometric study aims to analyze the global trends and scientific impact of topical probiotics in dermatology. By reviewing 106 articles published between 2013 and 2023, the study categorizes the applications of probiotics in wound healing, inflammatory skin diseases, and general skincare. The findings indicate a significant increase in publications from 2021 onwards, attributed to the heightened focus on medical research during the COVID-19 pandemic. This study also identifies the most productive countries, institutions, and authors in this field, highlighting the importance of international collaborations. The results underscore the efficacy of probiotic-based topical formulations in improving skin health, reducing inflammation, and enhancing wound healing. This comprehensive analysis supports the development of new therapeutic strategies based on topical probiotics and encourages high-quality research in this promising area.

## 1. Introduction

As the largest organ in the human body, the skin plays a multifaceted and crucial role in maintaining homeostasis. It serves as a vital physical barrier, regulating essential functions such as hydration, temperature control, and local immunity [1,2]. The skin shields the body from a variety of external aggressors, including pathogens, ultraviolet radiation, pollution, and harmful chemicals [3,4]. However, these environmental factors can promote the development of persistent skin disorders, such as dermatitis, acne, eczema, rosacea, psoriasis, wounds, and various infections. Such conditions can compromise the skin’s natural protective capabilities and adversely impact the quality of life for affected individuals [5,6].

Similar to the gastrointestinal tract, the skin harbors a complex community of microorganisms that interact with the host, protecting against pathogens, modulating the immune response, and preserving the skin barrier integrity [7,8]. The skin microbiome is a dynamic and complex ecosystem influenced by various factors, such as body region, age, gender, and geographic location. Common skin microbiota species include *Staphylococcus epidermidis*, *Cutibacterium acnes*, and *Corynebacterium* species, each playing distinct roles in maintaining skin health. For instance, *S. epidermidis* is known for its protective functions against pathogens [9], while *C. acnes* is involved in lipid modulation and can contribute to both skin health and acne development depending on its balance within the microbiome [10,11].

Studies have shown that the skin microbiome varies significantly across different areas of the body, such as the face, arms, and axillary region. In contrast, the arms and legs are commonly studied for conditions like eczema and psoriasis, where maintaining skin barrier function and hydration is crucial [12,13]. The axillary region, prone to body odor, is another area of interest, with research targeting *Corynebacterium* species to manage odor through probiotic interventions [9].

The use of probiotics and their derived metabolites (postbiotics) has emerged as a promising therapeutic avenue for a wide range of health conditions, including wound management, inflammatory diseases, and skin care [14,15,16]. Probiotics are particularly effective in treating skin diseases due to their ability to interact with the skin microbiota and modulate the local immune response [17,18]. These beneficial microorganisms can effectively colonize the skin, compete with pathogenic species, and exert a protective influence through various mechanisms, such as enhancing the skin barrier, reducing inflammation, and regulating sebum production [15,19]. Numerous studies have demonstrated the ability of probiotic strains to alleviate symptoms and improve clinical outcomes in conditions like acne, dermatitis, eczema, and psoriasis. By restoring the balance of the skin’s microbial community and dampening inflammatory pathways, probiotics offer a natural, non-invasive approach to managing a variety of cutaneous disorders [19].

The growing awareness of the importance of the skin microbiome has led to the introduction of various topical products targeting the skin microbial community. Several over-the-counter and prescription skin care formulations now incorporate probiotic strains such as lactobacilli and bifidobacteria [20,21,22], and even specific strains of *C. acnes* [10,11]. For example, the brand Aveeno offers a range of products, while La Roche-Posay’s Toleriane line incorporates postbiotic compounds to soothe sensitive skin. The growing availability of these topical probiotic-based solutions highlights the increasing recognition of the importance of maintaining a balanced skin microbiome for overall skin health and resilience. These topical probiotic-based solutions offer a more holistic and natural approach to managing skin disorders. The incorporation of these live microorganisms into skincare products highlights the evolving paradigm in dermatology, where maintaining a healthy skin microbiome is recognized as a crucial strategy for promoting overall skin health and resilience.

This bibliometric study aims to analyze the quantity and quality of scientific studies focused on the application of topical products based on probiotics (and/or their metabolites) and their impact on different skin conditions and body regions, especially in the face and arms. It seeks to highlight the main advantages and identify the challenges associated with this therapeutic approach. Additionally, this study aims to support the development of new therapeutic strategies based on topical probiotics and encourage high-quality research in this promising area.

This study did not cover systemic probiotics administered orally or through other routes. While acknowledging the influence of demographic factors on the skin microbiome, this study provided a general overview rather than a detailed analysis of these variations. The primary focus was the direct application of probiotics/postbiotic-based products to the skin and their potential benefits in dermatological treatments and skincare.

## 2. Materials and Methods

This study employed a bibliometric approach to map the scientific literature on the topical use of probiotics for skin applications, including healing processes, inflammatory diseases, and general skin care. The data search was conducted on the Web of Science platform on 28 November 2023.

The search was conducted using the following keywords combined with Boolean operators: (probiotic OR probiotics) AND (skin OR dermat) AND (topic), as can be seen in Figure 1. The analysis period spanned from 2013 to 2023, and only articles, reviews, and early access documents published in English were considered. In total, 203 articles were initially identified.

The authors conducted a thorough review of the titles and abstracts to include only studies focusing on the topical use of probiotics for skin applications, with a focus on healing processes, inflammatory diseases, and general skin care. Articles addressing the systemic absorption of probiotics were excluded as a criterion. After this screening process, 106 articles were selected, comprising 71 experimental studies and 35 review studies. The average number of citations per article within this database was then compared to a reference value (VREF) of 19.47 citations obtained from the Web of Science.

The studies were categorized based on their applications, which included wound healing effects, combating inflammatory skin diseases, and general skin care. Furthermore, the most commonly used probiotic strains and the types of formulations produced were identified and analyzed. Additionally, the annual publication trends were evaluated, and Welch’s *t*-test was applied to determine if there were significant changes in the number of publications over time.

Co-authorship analyses were conducted across countries, institutions, and authors, with rankings based on citation counts. The Shapiro–Wilk test verified the normality of the groups for the 10 most cited authors, followed by a one-sample *t*-test to compare the average citations per article against the reference value.

The analysis of citations between newspapers and articles included the identification of the 10 most cited newspapers, followed by the application of the Shapiro–Wilk normality test and the *t*-test for a sample, comparing the average number of citations received with the VREF. The 10 most cited experimental articles and the 10 most cited review articles were also listed.

The analysis also examined the percentage of publications in different Web of Science categories to identify the most prominent research areas. Additionally, the co-occurrence of keywords was mapped, and the bibliographic coupling between the 10 most cited experimental studies and the 10 most cited review articles was analyzed. The purpose of these analyses was to uncover the most extensively explored topics within the theme of topical probiotic use for skin applications.

## 3. Results

### 3.1. Annual Publications

The total number of publications on the topical use of probiotics from 2013 to 2023 and citations per year is represented in Figure 2. A significant increase was observed, particularly in 2021 and 2022, followed by a decline in 2023. To determine if the increase in publications from 2021 onwards was significant, we divided the publication data into two groups: 2013–2020 and 2021–2023. We then performed a Shapiro–Wilk normality test for each group, confirming that both followed a normal distribution. Subsequently, we applied Welch’s *t*-test, which is appropriate for samples with unequal variances. The analysis revealed a *p*-value of 0.0360, indicating a statistically significant difference between the groups. Specifically, the mean number of publications in 2021–2023 was significantly higher than in 2013–2020, with a mean difference of 17.46 publications. Additionally, the F-test performed to compare variances confirmed that the variances were also significantly different between the groups. Collectively, these results confirm that the increase in publications from 2021 onwards was statistically significant.

This can be attributed to the COVID-19 pandemic, which stimulated a general rise in scientific production in life sciences and biomedicine, as reported by recent studies [23]. During the pandemic, there was an increased focus on medical research and the exploration of new therapies, leading to more experimental and review publications. The drop observed in 2023 can be explained by the end of the pandemic period and a possible normalization of research activities, resulting in a stabilization in the rate of new publications. The increasing citation curve over the years reflects the continued relevance and impact of this research in the field of dermatology.

### 3.2. Most Productive Countries, Organizations, and Authors

In this section, we present the ranking of the most influential countries, organizations, and authors in studies on the topical use of probiotics on the skin. For this purpose, we used the number of citations that each study received.

#### 3.2.1. Most Productive and Influential Countries

The experimental studies on topical probiotics on the skin were published by 33 countries. The most extensive set of countries connected in a co-authored analysis in the collaboration network consisted of 12 countries. At this point, we did not apply exclusion criteria regarding the number of publications and citations.

Figure 3 presents the mapping of co-authorship between countries. The country co-authorship map illustrates the primary collaborations and geographic distribution of research on the topical application of probiotics for wound healing, inflammatory processes, and skin care. The node sizes, representing publication counts, indicate that the United States and France lead in research output, with the largest nodes and numerous international connections.

The group with the highest number of connections includes a core of 12 collaborating countries, such as Switzerland, Malta, Italy, India, Taiwan, Singapore, Canada, Brazil, Mauritius, and Nigeria, in addition to the United States and France. This diverse international collaboration suggests a well-established network of research and knowledge exchange, which is crucial for advancing scientific progress.

Furthermore, among the 33 active countries, 18 have published at least two works, 11 have published at least three, 9 have published at least four, and 5 countries have published at least five works. This demonstrates consistent dedication across multiple nations, with a smaller but significant number of highly productive countries making substantial contributions to the field. The multiple connections between nodes suggest that research on topical probiotics is a globally recognized area with strong potential for impact due to the diversity of perspectives and shared knowledge.

This study investigated the temporal trends in international research collaborations focusing on the topical application of probiotics for wound healing, inflammatory processes, and skin care. Early publications from 2019 to 2020 indicate that Switzerland and the United States were partnering on this research. By 2021, the analysis identified increased collaborative efforts between France, Italy, and India. More recent findings suggest that Taiwan and Singapore have also joined the collaborative network, signaling ongoing growth and interest in this research area.

International collaborations have produced promising results in the development and application of topical probiotics for skin care. These studies have contributed to advancements in both scientific knowledge and practical applications in this field. In the red cluster, a study involving researchers from Nigeria and Canada [9] created a topical cream containing *Lactobacillus pentosus* KCA1 (now classified as *Lactiplantibacillus pentosus*) to reduce bacteria that cause unpleasant body odor in healthy individuals. The clinical trial of 25 volunteers showed decreases in the frequencies of Actinobacteria, *Corynebacterium*, and Firmicutes, while an increase in lactobacilli was observed. Additionally, the bacterial genes linked to odor production were downregulated, leading to reduced body odor. This demonstrates the efficacy of topical probiotics in modulating the skin microbiome and improving quality of life through body odor control.

A collaboration between researchers from Canada, Brazil, Mauritius, and France (green cluster) developed the Mineral 89 Probiotic Fractions, a formulation to reduce retinoid-induced irritation and improve aging skin health. A clinical study with 38 women found that using this formulation alongside tretinoin improved skin hydration, reduced inflammation, and increased participants’ quality of life, highlighting the potential of probiotics in cosmetic dermatology [24].

Within the green cluster, researchers from Switzerland and France conducted two studies using *Lactobacillus johnsonii* NCC 533 [25,26]. In the first in vitro study, the heat-treated strain reduced *Staphylococcus aureus* adhesion and increased antimicrobial peptide expression in a reconstructed human epidermis model. In the second clinical trial on patients with atopic dermatitis, a lotion containing this heat-treated strain (parabiotic formulation) reduced *S. aureus* burden in lesional skin and improved the SCORAD score. These results demonstrate the effectiveness of topical probiotics in strengthening skin immunity and reducing harmful bacterial colonization.

A partnership between Malta and France within a green cluster [27] focused on using *Lactiplantibacillus plantarum* to treat acne. In an in vitro study, this probiotic formulation reduced the adherence of *S. aureus* bacteria and stimulated the production of antimicrobial peptides, suggesting its potential as a complementary therapy for acne. These results highlight the ability of probiotics to modulate the skin’s microbial community and strengthen the skin’s natural defenses against pathogens.

A collaboration between researchers in France (green cluster) and countries belonging to the blue cluster (USA and Italy) [28] examined the effects of M89PF, a topical postbiotic formulation containing Vichy mineralizing water and probiotic components, on preventing UV- and ozone-induced skin damage. In vitro experiments revealed that M89PF effectively protected the skin barrier from degradation and mitigated oxidative and inflammatory stress markers, demonstrating its potential to shield skin from harmful environmental insults. This work underscores the significance of incorporating topical probiotics to safeguard and preserve the integrity of skin exposed to environmental stressors.

An India–USA study (blue cluster) [29] examined the protective effects of LactoSporin, a formulation containing the extracellular metabolite from *Bacillus coagulans*, against UV- and ozone-induced skin damage. Laboratory findings showed that LactoSporin prevented the reduction in skin barrier markers and the induction of oxidative and inflammatory stress, indicating its potential as a skin protective agent.

A collaborative research study involving scientists from the United States (blue cluster) and Taiwan and Singapore (yellow cluster) [30] investigated the effects of butyric acid, a metabolite (postbiotic) produced by the probiotic bacterium *S. epidermidis*, on skin physiology. Using an in vivo experimental model, the researchers determined that topical application of butyric acid reduced the UVB-induced production of the inflammatory cytokine IL-6, and this effect was mediated through the short-chain fatty acid receptor FFAR2. This study provides mechanistic insights into how probiotic metabolites can modulate the inflammatory response in the skin, suggesting a novel therapeutic approach for the management of various skin inflammatory conditions.

Table 1 presents the 10 countries with the highest number of citations received. The citation data for countries that have published research on the topical application of probiotics provide insights into the influence and impact of each nation’s contributions. The United States emerges as a leader in this field, with 13 published articles and 108 citations, reflecting its strong performance and influence. Its substantial total link strength indicates extensive international collaboration. France also stands out, with 7 publications and 94 citations, distinguished by a high average citation per article, suggesting the high quality and relevance of its work. Its significant total connection strength demonstrates a robust collaborative network.

Other nations, such as South Korea and Sweden, have fewer international connections but exhibit high average citations per article, 15.8 and 23.67, respectively. This indicates the impact of their independent research. Iran has one of the highest averages of 33.5 citations per article, suggesting that its fewer contributions are highly impactful. Switzerland, with 2 publications and 66 citations, shows robust results and a high average citation per article. Portugal and Italy also have significant influence, with averages of 20.3 and 15 citations per article, respectively.

Singapore and Taiwan, each with one highly cited publication, demonstrate the importance of their international collaborations. Japan, with 2 publications and 48 citations, stands out for the quality of its research and effective international collaborations, reflected in a high average citation per article.

Overall, the data show that the United States and France lead in the volume and impact of publications, reinforcing their positions as influential research centers. Other countries, despite fewer publications, have very high average citations per article, highlighting the exceptional quality of their work. Several nations demonstrate strong international collaborations and high-impact publications, contributing significantly to advancing knowledge in wound healing, inflammatory processes, and skin care.

The statistical analysis indicates that the top 10 most cited countries exhibit a higher average citation rate per article compared to the overall average of 14.4 on the Web of Science database. The mean citation count was 23.60, differing by 9.204 from the expected value. The t-statistic was 2.408, with nine degrees of freedom, yielding a *p*-value of 0.0394. This suggests that the difference between the countries’ mean citations and the reference value is statistically significant at the 0.05 level.

The 95% confidence interval for the discrepancy ranged from 0.5584 to 17.85, suggesting that the countries’ average citations are significantly higher than the reference value. Approximately 39.19% of the variance in the citation averages can be explained by the difference between the observed values and the reference value, as indicated by the R-squared value of 0.3919. The analysis suggests that the countries involved display notably higher average citation counts compared to the reference value of 14.4, implying their significant influence and impact within this field of research.

#### 3.2.2. Most Productive and Influential Organizations

The investigation of institutional co-authorship revealed a network of 131 organizations conducting experimental research on the topical application of probiotics. However, only 18 institutions had published at least two articles, implying that most contribute fewer publications. Figure 4 presents the 19 most collaborative institutions, which serve as important interaction hubs, suggesting that strategic partnerships are crucial for advancing topical probiotic research. The size of the nodes, which represents the number of publications, accentuates the most productive and collaborative institutions, providing a detailed perspective on significant interactions and partnerships. Within this landscape, the study identified four prominent collaboration groups with the strongest connections between institutions.

The blue group, comprising Greek institutions like Uni Pharma SA, Aristotle University of Thessaloniki, Democritus University of Thrace, and the University of Crete, has seen a recent surge in collaborative publications between 2021 and 2022. This reflects a strong synergy between universities and pharmaceutical companies in Greece, advancing topical probiotic research.

The yellow group, with institutions such as Nestlé Skin Health Galderma Research, Nestlé Research Center, and University Hospital St. Etienne, has maintained ongoing collaborations since 2018. This stable, long-term partnership suggests a robust research network dedicated to dermatology and the scientific investigation and clinical application of probiotics.

The green group, made up of organizations like L’Oréal Paris, L’Oréal Brazil, Vichy Labs, and Windsor Clinical Research Inc., has established a network over several years, with a focus on publications between 2020 and 2022. The prominence of cosmetic companies indicates the continued integration of scientific research and commercial product development in the skin care sector. The purple cluster, represented by Beijing Technology and Business University and Beijing Key Laboratory of Plant Resources & Dev., features recent collaborations between 2022 and 2023.

Finally, the red group comprises South Korean universities, including Kyung Hee University, Sungkyunkwan University, and Kangwon National University, exhibiting intense academic collaboration, with recent publications mainly between 2021 and 2023, highlighting the strong network of academic partnerships in South Korea and their significant contribution to the advancement of topical probiotic research.

Institutional collaborations have driven crucial advancements in topical probiotic research. Two pivotal studies within the blue cluster investigated the application of probiotics, including *L. plantarum*, *Lacticaseibacillus rhamnosus*, and *Bifidobacterium longum*, to promote healing in excisional wounds of Wistar rats [21,31]. These studies analyzed how probiotics expedite healing and influence inflammatory, reparative, and angiogenic factors. In vivo findings revealed that probiotics significantly accelerate the healing process, with varying strains demonstrating greater efficacy at different stages. This underscores the need for a combined probiotic formulation to optimize the healing outcome.

Researchers in the yellow cluster investigated the effects of the probiotic *L. johnsonii* NCC 533 on reducing *S. aureus* adhesion and stimulating antimicrobial peptide production [25,26]. This probiotic was evaluated in laboratory models of human skin as well as in patients diagnosed with atopic dermatitis. Both in vitro and clinical studies demonstrated that topical application of this probiotic significantly diminished *S. aureus* colonization and bolstered the skin’s innate immune defenses. Clinical trials also showed that probiotic use improved patients’ SCORAD scores, suggesting a reduction in the severity of atopic dermatitis.

Within the green cluster, the studies investigated the impacts of products containing probiotics, Vichy volcanic water, hyaluronic acid, niacinamide, and tocopherol, utilized in combination with topical retinoids to address photoaging [24,28]. In a clinical trial involving 38 female participants who used the M89PF formulation for 84 days, significant improvements were observed in skin hydration, reduced irritation from retinoids, and enhanced appearance of fine lines, skin tone, radiance, and pore size. Furthermore, the formulation was associated with decreased markers of inflammation and oxidative stress, indicating good tolerability and potential anti-aging benefits.

In a study by researchers from the red cluster, a fermented Korean red ginseng extract containing yeast and probiotics was evaluated for its effects on 1-chloro-2,4-dinitrobenzene-induced allergic skin inflammation in Balb/c mice [32]. Topical application of the extract was found to mitigate scratching behavior and reduce serum levels of IgE and IL-4. Additionally, the treatment suppressed the phosphorylation of p38 MAPK and activation of NF-kappa B, indicating the extract’s potential as a therapeutic agent for atopic dermatitis.

Finally, the two studies within the purple cluster highlight the positive impact of Bifidobacterium-derived probiotics on skin barrier function. Bifida Ferment Lysate was found to improve skin barrier strength and reduce oxidative stress, while Bifidobacterium infantis extract demonstrated strong antioxidant and barrier-strengthening properties, and is safe for topical use. These studies reinforce the potential of probiotics in topical treatments to enhance skin health, particularly in conditions like atopic dermatitis [33,34].

The collaborative research has yielded crucial findings, ranging from enhancing wound repair to mitigating bacterial proliferation in skin disorders, as well as offering anti-aging cosmetic advantages and managing allergic skin inflammation. Laboratory investigations, cell-based studies, and clinical trials have substantiated the significance of combined probiotic formulations and their potential therapeutic and cosmetic applications. These advancements promote skin wellness and patient well-being, underscoring the importance of international and multidisciplinary scientific collaborations.

Table 2 presents a ranking of institutions based on their citation counts for experimental articles exploring the topical use of probiotics. This ranking provides valuable insights into the influence and impact of each institution within the research field. The table emphasizes the collaborative nature of scientific endeavors by highlighting both individual institutions and research consortia.

Shiraz University has published two impactful papers that have collectively garnered 67 citations, equating to an average of 33.5 citations per publication. This suggests that the university’s research is highly esteemed within the scientific community, despite a lack of extensive international partnerships.

Nestlé Research Center and Nestlé Skin Health Galderma R&D have exhibited exceptional research productivity, generating two publications that have collectively garnered a significant number of citations. The network of international collaborations demonstrates the leadership and influence of these organizations within the field. The research collaboration between Malmö University and BioGaia AB has been noteworthy, with their two joint publications receiving an average of 31.5 citations per article, which showcases the effectiveness of their international research partnership. This collaboration highlights the significance and quality of the research conducted by these institutions.

The Portuguese Catholic University has a relatively limited number of international research collaborations, but the high average of 20.33 citations per article across its three publications suggests the strong quality and impact of its internal research program. Yeungnam University, with an average of 29.5 citations per article and a considerable number of international collaborations [total link strength (STL) = 4], stands out for its significant contributions to topical probiotic research.

This collaborative research between University Hospital of Saint-Etienne, Galderma Int, and Mucosal Immune & Pathogen Agents Group produced a highly influential article, with an average of 50 citations, highlighting the significance of the study. This internationally collaborative article, involving researchers from National Central University, National University of Singapore, and University of California San Diego, has been cited 49 times, underscoring the impact of this cross-border research endeavor.

The independent research conducted at Manchester University has had a substantial impact, with an average of 22 citations per article across two publications. This widely cited publication, which has been referenced 43 times, was produced through collaboration between the University of Eastern Piedmont, Verona University, Oriflame Cosmetics, and Sintal Dietetics S.R.L., highlighting the importance of public–private partnerships in academic research.

The collaborative research between Universiti Sains Malaysia, Inha University, Dongguk University, Perdana University, the Ministry of Science ICT and Future Planning, and RIKEN led to a highly influential publication, with 36 citations, underscoring the significance of joint academic and governmental efforts.

A parametric *t*-test was employed to evaluate the average number of citations per article for the 10 most frequently cited organizations or groups of organizations, comparing this metric to the reference value of 14.4. This analysis was intended to offer insights into the statistical significance of these comparative citation patterns. The statistical analysis reveals a significant difference between the average citations per article for the 10 most cited organizations and the reference value, as evidenced by the low *p*-value.

The *t*-test compared the average number of citations for the institutions, which was 34.78, to the reference value of 14.4. This showed a difference of 20.38 between the observed and theoretical values. The statistical analysis yielded a t-value of 6.380, with nine degrees of freedom, resulting in a *p*-value of 0.0001. This *p*-value indicates that the difference between the institutions’ average citations and the reference value is statistically significant at a 0.05 significance level.

The 95% confidence interval for the difference ranged from 13.16 to 27.61, suggesting that the institutions’ average citations were meaningfully higher than the reference value. The R-squared value of 0.8190 demonstrated that approximately 82% of the variation in the citation averages could be attributed to the divergence between the observed values and the reference value.

The institutions have achieved a notably higher average number of citations than the reference value of 14.4, indicating their significant impact in the field.

#### 3.2.3. Most Productive and Influential Authors

This examination of the collaborative author network for experimental publications on the use of probiotics in wound healing, skin inflammation, and general skin care offers insights into the nature of these research collaborations and their development over time. The network comprises 418 authors, but only 28 have co-authored two or more articles. For the visual representations in Figure 5, the 20 authors with the most substantial connections were selected.

The co-authorship network analysis reveals distinct clusters with many degrees of collaboration among the researchers. The yellow cluster encompasses authors such as A. Gueniche and D. Kerob, whose close partnership suggests a focus on specific studies concerning topical probiotics. This cluster demonstrates recent collaborative research, predominantly from 2022 to 2023, suggesting that these authors have consistently contributed to the recent literature. Conversely, the green cluster, including authors like Y. Pan, J. Zhao, and Y. Han, exhibits a high level of intense collaboration with numerous connections between the researchers, indicative of an active and cohesive research group. Collaborations in the green cluster are recent, with a focus on publications from 2022 to 2023, suggesting that this research group is active and expanding, continuously publishing new findings.

The red cluster comprises authors including *G. Stavrou*, *E. Filidou*, *G. Kolious*, and *K. Kotzampassi*. This group exhibits a high degree of interconnectedness, indicating robust and fruitful collaboration among its members. This cluster represents a combination of older and newer collaborative publications, indicating the continuity and development of research in this area. Conversely, the blue cluster, which features authors such as A. S. Monteiro, C. A. Monteiro, and L. C. N. da Silva, demonstrates significant collaborations, though they are less dense compared to other clusters. This cluster indicates recent collaborations, as shown in published works, suggesting that new areas of research or partnerships have emerged.

The study authors developed several probiotic-based topical formulations for wound healing, skin infection control, and general skin care. The blue cluster formulated gels with hydroxyethylcellulose and alginate containing Lactobacilli strains, evaluating their antimicrobial properties. In vitro and ex vivo experiments showed that these formulations effectively inhibited pathogens like *S. aureus* and Pseudomonas aeruginosa, highlighting the efficacy of *L. plantarum* and the formulations’ stability under various storage conditions [35,36].

Researchers in one study group developed the Mineral 89 Probiotic Fractions formulation. This combined probiotic fractions with Vichy volcanic mineralizing water and other components. The goal was to reduce irritation caused by tretinoin and improve skin hydration and appearance. A clinical trial with 38 women showed that this formulation not only reduced irritation but also improved participants’ quality of life, suggesting its potential as a supplemental product for treating photoaging [24,37].

The studies in the red cluster examined the topical use of various probiotic strains, including *L. plantarum*, *L. rhamnosus*, and *B. longum*, in rat wound models. These animal studies found that probiotic formulations sped up the wound healing process, with different strains being more effective at specific stages. Additionally, combinations of multiple probiotic strains were more effective at promoting healing and angiogenesis, suggesting that combined probiotic treatments are important for modulating the different phases of the healing process [21,31].

The green cluster of studies focused on the antioxidant and skin-barrier-strengthening effects of Bifidobacterium yeast extracts and *B. infantis* culture supernatants (postbiotics). In laboratory tests, these formulations enhanced the expression of genes linked to the skin’s physical barrier and antimicrobial peptides, and also exhibited strong antioxidant properties. Additionally, these formulations were found to be safe for topical use, indicating their potential to improve the skin’s resistance to oxidative stress and inflammation [33,34].

Table 3 presents the 10 authors most frequently cited in experimental research on the topical use of probiotics for wound healing, skin inflammation, and skin care. The table includes the number of articles that each author published, their total citations, citations excluding self-citations, citations per article, and total link strength. This highlights the importance of these authors in the field of studying the topical application of probiotics for wound healing, skin inflammations, and general skin care.

The top-cited researchers are A. Mercenier and E. Butler, with 66 and 63 citations, respectively, and averaging 33 and 31.5 citations per article. Dr. A. Mercenier also has a high collaboration score of 12, indicating strong collaborative work. F. K. Tavaria has 61 citations, or 60 excluding self-citations, averaging 20.33 citations per article and a collaboration score of 8. Y.-H. Park and A. Oryan are also highly cited, with 59 and 51 citations, respectively, with A. Oryan’s 51 citations per article being particularly notable.

S. Blanchet-Rethore and C.-M. Huang have high citation counts of 50 and 49 citations per article, respectively. C. A. O’Neill, D. Cecconi, and J. M. Delgado-Lopez have lower citation counts, ranging from 44 to 25 citations, and 22 to 14 citations per article. The differences in total citations and citations per article indicate varying levels of collaboration among these authors in the field.

Removing self-citations did not greatly change the ranking, but gave a more accurate view of each author’s real influence. The average citations per paper varied greatly across authors, showing the diversity of impact in their published work. The total strength of the connections indicates the network of collaborations and connectivity among these authors in the research area.

We conducted an analysis comparing the average citation count of the 10 most cited corresponding authors, which was 29.90, to a reference value of 14.4. This represents a difference of 15.50 between the current average and the reference value. The statistical test yielded a t-value of 3.015 with nine degrees of freedom and a *p*-value of 0.0146. This *p*-value indicates that the difference between the authors’ average citations and the reference value is statistically significant at the 0.05 significance level.

The 95% confidence interval for the discrepancy was between 3.871 and 27.13, suggesting that the average authors’ citations were significantly higher than the reference value. The R-squared value was 0.5025, indicating that approximately 50% of the variance in the citation averages could be explained by the difference between the observed and reference values.

The results suggest that the most cited corresponding authors have significantly more citations on average than the reference value of 14.4, indicating that these authors have a greater impact in the field.

### 3.3. Most Active Journals in the Field

The analysis revealed that 106 publications focused on topical use of probiotics in wound healing processes, inflammatory processes, and general skin care. Among these, ten journals received more citations. Table 4 presents the 10 most cited journals.

The journal ‘Experimental Dermatology’ is the most cited, accumulating 67 citations from four articles, resulting in an average of 16.75 citations per article and a total link strength (STL) of 16, highlighting its influence on the network of quotes. Next, the ‘Journal of Applied Microbiology’ occupies second place, with 55 citations for one article, giving an average of 55 citations per article and an STL of 4. The journal ‘Burns’ also stands out, with 51 citations for a single article, resulting in an average of 51 citations per article and an STL of 1.

Another prominent journal, ‘Clinical Cosmetic and Investigational Dermatology’, has 50 citations across four articles, resulting in an average of 12.5 citations per article and an STL of 3. The ‘International Journal of Molecular Sciences’ has 49 citations for one article, with an average of 49 citations per article and an STL of 1. ‘Probiotics and Antimicrobial Proteins’ has 43 citations for two articles, with an average of 21.5 citations per article and an STL of 10, while ‘Scientific Reports’ has 43 citations for two articles, with an average of 21.5 citations per article and an SLT of 7.

‘Applied and Environmental Microbiology’ features a single publication with 41 citations, reflecting an average citation count of 41 per article and an STL of 1. The journal ‘Microorganisms’ hosts 39 citations across four articles, equating to an average of 9.75 citations per publication and an STL of 6. ‘Frontiers in Microbiology’, on the other hand, records 36 citations for a single article, averaging 36 citations per publication and an STL of 1.

The average number of citations per article from the top 10 most highly cited publications is 31.40, substantially higher than the reference value of 14.4. The statistical analysis revealed a t-value of 3.162 with nine degrees of freedom, and a *p*-value of 0.0115, which is below the 0.05 significance threshold. This finding indicates that the observed difference is statistically significant at the 5% level. Additionally, the 95% confidence interval for the disparity ranges from 4.839 to 29.16, suggesting that the average citation count per article is notably higher than the reference value.

The average discrepancy was 17.00, with a standard deviation of 17.00 and a standard error of the mean of 5.376. The R-squared value of 0.5263 indicates a moderate to large effect on the variation in the average number of citations per article relative to the reference value.

The analysis of articles in the top journals showed various topical probiotic products for treating skin conditions. These studies examined the antimicrobial, anti-inflammatory, and skin-barrier-enhancing properties of probiotic-based gels, creams, ointments, and lotions. Figure 6 presents the citation mapping network, illustrating the total link strength in relation to citations between the 10 most highly cited publications.

The green cluster of articles focuses on probiotics like *Limosilactobacillus reuteri*, *L. johnsonii*, and *B. infantis*. These probiotic formulations, including creams and lotions, have shown effectiveness in reducing skin inflammation, strengthening the skin barrier, and having antioxidant properties. Studies suggest that these probiotics may help with conditions such as atopic dermatitis and acne, as well as improving overall skin health [9,11,19,25,27,33,34,38,39,40,41,42,43,44,45].

The studies in the red cluster examine the topical use of probiotic strains like *L. plantarum*, *L. rhamnosus*, and *B. longum*. These investigations evaluated the efficacy of gel, cream, and lotion formulations containing these probiotics in in vivo and in vitro models. The research focused on the potential of these probiotic-based products to promote wound healing, reduce inflammation, and inhibit skin pathogens. The findings suggest that these formulations can accelerate the wound healing process and enhance the skin’s immune response [18,21,31,46,47,48,49,50].

The articles in the blue cluster focus on cellulose-based gels and alginate containing the probiotic strains *L. plantarum* and *L. rhamnosus*. These gels showed antimicrobial effectiveness against *S. aureus* and *P. aeruginosa* in ex vivo experiments, suggesting their potential for developing new wound dressings to treat infected injuries. Additionally, articles in the Journal of Applied Microbiology examined the adhesion, antimicrobial activity, and biofilm formation of various probiotic strains in in vitro studies. These findings highlight the potential of probiotics as a complementary treatment for skin disorders, promoting a healthy skin microbiome [35,36,51].

### 3.4. Highly Cited Papers

We identified 106 articles published in the analyzed period on the topical use of probiotics on the skin. Thirty-five studies are classified as review articles, representing 33.02%. The other 71 studies are experimental articles. This subdivision can be seen in Figure 7. Twenty-five articles were identified as proposals for systems for wound healing, forty-five for inflammatory skin diseases, and thirty-six for skin care in general.

Table 5 and Table 6 present the 10 most cited experimental and review papers from the analyzed period.

### 3.5. Web of Science Categories

The 106 articles analyzed in the research are distributed across 30 Web of Science categories. Dermatology has the highest percentage of publications, with 39.62% (42 publications), followed by Microbiology (20.76%; 22 publications) and Pharmacology/Pharmacy (13.21%; 14 publications). Figure 8 shows the 10 Web of Science categories with the most articles included.

### 3.6. Terms Occurrence Network Analysis

Analysis of keywords in experimental articles on the topical use of probiotics on the skin provided valuable insights into central themes and connections between different research areas. A total of 444 keywords were identified and the network of keyword co-occurrences was mapped based on a minimum of five occurrences for each keyword. Figure 9 presents the mapping of keywords with at least five occurrences, totaling 16 words.

The key terms and concepts identified in the research network include “probiotics”, “skin”, “microbiome”, “atopic dermatitis”, “inflammation”, “wound healing”, “prebiotics”, and “in-vitro”. These terms highlight the main research areas, with “probiotics” being the central and most commonly used keyword, indicating the primary focus of the studies.

The text indicates a strong focus on the direct application of probiotics to the skin, reflecting the importance of the skin microbiome. There is a considerable emphasis on investigating the effects of probiotics on inflammatory skin conditions, such as atopic dermatitis, as well as their use in wound healing processes. The text also suggests that studies examine the skin health benefits of prebiotics in addition to probiotics. Additionally, the prevalence of in vitro methods used to investigate the effects of probiotics is noted.

The identified research clusters reveal a diverse range of focus areas. The red cluster highlights terms like “probiotics”, “skin”, “acne”, “bacteria”, and “diversity”, suggesting a focus on skin conditions, acne treatments, and microbial diversity. The green cluster includes words such as “atopic dermatitis”, “inflammation”, “disease”, and “skin barrier”, indicating research on atopic dermatitis, inflammation, and skin barrier function. The blue cluster focuses on “wound healing”, “bacteria”, “probiotics”, “prebiotics”, and “laboratory studies”, reflecting research on wound healing, specific probiotic strains, and prebiotics.

The studies examined various aspects of skin health, including inflammatory conditions, microbial composition, and wound healing. The central theme is the strong connection between “probiotics” and “skin”. The focus on inflammatory conditions is highlighted by the link between “atopic dermatitis” and “inflammation”, while the importance of the skin’s microbial composition is indicated by the intersection of “microbiome” with several keywords. Additionally, the connection of “wound healing” with “in-vitro” suggests that many of the studies on wound healing were conducted in controlled laboratory settings.

### 3.7. Analysis of the Most Used Probiotic Species

Figure 10 provides a comprehensive overview of the probiotic species most commonly applied in the enrolled studies. The *L. plantarum* strains were the most applied across all study types, particularly in in vitro, in vivo, and clinical studies, followed by *L. rhamnosus* and *Lacticaseibacillus casei*. *B. longum* and *Limosilactobacillus fermentum* share the fourth place, with *B. longum* strains being more used in in vivo studies, while *L. fermentum* strains were most employed in in vitro studies. Interestingly, *L. fermentum* and *B. longum* strains were not applied in clinical studies enrolled in this study.

### 3.8. Analysis of the Types of Formulations Used in the Studies

The heatmap table in Figure 11 categorizes probiotics-based formulations developed for topical use according to the type of study that they were evaluated as (in vitro, ex vivo, in vivo, and clinical studies). Creams emerged as the most extensively studied formulation, with 13 studies, particularly in clinical settings (7 studies). This indicates a strong interest and potential efficacy in using creams for topical probiotic applications. Gels and ointments also showed a notable number of studies (seven and six, respectively), with four in vivo studies each.

Serums and lotions, although having fewer studies overall, still showed some clinical and in vivo research, indicating ongoing interest in their potential benefits for skin care. Dermocosmetics and emulsions had a mix of study types, with a few clinical studies, gaining attention for their dual role in skincare and therapeutic benefits. Films, emollients, pads, and baths had the least number of studies, suggesting that they are less commonly used or newer areas of research in the field of topical probiotics.

### 3.9. Analyzing the Bibliographic Connections between Experimental Studies of Topical Probiotics

Figure 12 shows the bibliographical coupling between the 10 most cited articles. Analysis of the 10 most influential papers on topical probiotic applications for skin care reveals diverse formulations, applications, and study types, suggesting the potential benefits of probiotics in dermatology.

In the blue cluster, the article by Rather et al. [20] found that an ethanol extract from the probiotic *Latilactobacillus sakei* proBio-65 protected against imiquimod-induced skin inflammation resembling psoriasis in mice. The results showed that this probiotic treatment significantly reduced epidermal thickness, inflammatory cell infiltration, and pro-inflammatory cytokine expression, suggesting its potential as a psoriasis treatment. In the referenced study, Park et al. [54] investigated the comparative effects of an emollient containing *L. sakei* probiotic 65 and a standard emollient in patients diagnosed with atopic dermatitis. The randomized, double-blind clinical trial revealed that the probiotic-infused emollient enhanced skin permeability, diminishing transepidermal water loss and augmenting skin hydration levels.

In the green cluster, Lopes et al. [51] evaluated several probiotic strains in in vitro tests, focusing on their ability to adhere, inhibit microbes, and prevent biofilm formation. Most of these strains showed substantial antimicrobial activity and successfully prevented biofilm development, suggesting their potential as supplements to traditional treatments for skin conditions. The research of Oryan et al. [22] examined the effects of collagen hydrogel, both with and without the addition of Saccharomyces cerevisiae, on the healing of burn wounds in rats. The combined treatment was observed to enhance the healing process, epithelialization, and biomechanical properties of the wounded areas, producing comparable results to the application of silver sulfadiazine. The study of Mohammedsaeed et al. [39] strongly suggested that the probiotic *L. rhamnosus* GG can protect skin cells from *S. aureus* infection. The study found that this probiotic increased cell survival and reduced bacterial attachment based on laboratory experiments.

The red cluster highlighted research by Brandi et al. [18], who investigated the wound-healing, anti-inflammatory, antimicrobial, and proteomic properties of Lactobacillus lysates (postbiotic). These findings suggest that these probiotics may be utilized in combination with standard treatments to mitigate diverse skin conditions. In a multicenter exploratory study, Blanchet-Réthoré et al. [26] found that a lotion containing heat-treated *L. johnsonii* NCC 533 reduced *S. aureus* levels and improved atopic dermatitis symptoms. The research of Khmaladze et al. [19] explored the effects of topical applications of both viable and lysed *L. reuteri* DSM 17938. The results indicated that both forms possessed anti-inflammatory capabilities and improved skin barrier function while the viable form also exhibited antimicrobial activity. The study of Keshari and collaborators [30] found that butyric acid (postbiotic) produced by the probiotic *S. epidermidis* can reduce the production of inflammatory cytokines caused by UVB exposure in an experimental mouse model. Isolated from other clusters, Ong et al. [48] investigated the anti-staphylococcal effects of *L. plantarum* USM8613, demonstrating its ability to promote wound healing and innate immunity, as observed through laboratory and animal studies.

The reviewed literature demonstrates the wide-ranging approaches and promising efficacy of probiotics in dermatological applications, spanning wound healing and the management of chronic inflammatory skin disorders like atopic dermatitis and psoriasis. The robust bibliographic connections among these publications underscore the relevance and interdependence of research in this domain.

### 3.10. Analyzing the Bibliometric Connections between Review Studies on Topical Probiotics

Figure 13 shows the top 10 cited review articles that provide important insights into using topical probiotics for skin health. One article in the red cluster, by Sikorska and Smoragiewicz [54], explores how probiotics can prevent and treat methicillin-resistant *S. aureus* infections, highlighting the effectiveness of certain *lactobacilli* and *bifidobacteria* strains against clinical MRSA. Another review, by Al-Ghazzewi and Tester [16], discusses the impact of prebiotics and probiotics on skin health, emphasizing their ability to optimize, maintain, and restore the skin microbiome through both topical and systemic application. Yu et al. [53] conduct research on the microbiome’s role in dermatology, evaluating probiotic treatments for conditions like atopic dermatitis, acne, and psoriasis. Wollina [56] explores the skin microbiota in atopic dermatitis, noting increased *S. aureus* and decreased diversity during outbreaks. Rozas et al. [10] examine the role of *C. acnes* in maintaining healthy skin, considering its potential as a dermal probiotic. Finally, the review of Notay et al. [57] highlights the promising use of probiotic, prebiotic, and synbiotic supplements for preventing and treating skin conditions in adults, particularly atopic dermatitis, and as an additional treatment for acne.

In the green cluster, the manuscript from Franca [58] examines the current scientific literature on the topical use of probiotics and their impacts on dermatological conditions and skin health, investigating whether they can provide comparable benefits to orally administered probiotics. Lew and Liong [55] examine the mechanisms and compounds of probiotics that promote skin health, highlighting their benefits for improving conditions like eczema, dermatitis, and the healing of burns and scars. Lew also discusses how probiotics can rejuvenate the skin and enhance its natural immunity. Knackstedt et al. [14] analyze how skin microbiota impacts wound healing and focus on the therapeutic potential of topical probiotics to prevent infections, regulate inflammation, and improve wound healing. Additionally, Knackstedt and collaborators [15] study the role of topical probiotics in treating skin conditions, emphasizing the need for further research on their potential to address dysbiosis and other skin issues.

## 4. Discussion

This bibliometric study provides a comprehensive overview of the scientific literature on the topical use of probiotics for wound healing, inflammatory skin diseases, and general skin care. The enrolled studies, in particular the clinical trials, highlight the beneficial effects of probiotics on skin health by improving skin barrier function, reducing inflammation, and enhancing wound healing [19,22,25].

The beneficial effects of probiotics on the skin can be attributed to several mechanisms. Probiotics modulate the skin microbiome, enhancing the growth of beneficial bacteria while inhibiting pathogenic species [7,8]. They also exert anti-inflammatory effects by downregulating pro-inflammatory cytokines and upregulating anti-inflammatory mediators [17,18]. Additionally, probiotics strengthen the skin barrier by promoting the production of ceramides and other essential lipids [3,4]. These mechanisms collectively contribute to the maintenance of skin homeostasis and the prevention of various dermatological conditions.

Among the probiotic strains analyzed in this review, *L. plantarum* stands out for its broad applicability in different dermatological conditions. Studies have shown that this strain not only improves skin barrier function by reducing transepidermal water loss (TEWL) but also combats skin inflammations and infections by inhibiting the growth of pathogens such as *S. aureus* and *P. aeruginosa* [32,44]. Additionally, an acceleration in the wound healing process was observed in wounds treated with *L. plantarum*, highlighting its potential in skin regeneration [48,59].

The strain *L. rhamnosus* has proven to be particularly effective in treating inflammatory skin diseases, such as atopic dermatitis, where topical application significantly reduced inflammation levels while restoring the balance of the skin microbiome [39]. This strain has also been associated with improved skin hydration, making it a promising candidate for managing inflammatory conditions and dryness [41,59].

*L. casei*, in turn, has demonstrated efficacy in promoting wound healing and reducing signs of inflammation in experimental models, as well as showing potential to balance the skin microbiota, especially in inflammatory conditions such as acne and dermatitis [32]. Together, these three strains present a wide range of dermatological benefits, from wound healing to reducing inflammation and improving hydration, making them highly relevant for the development of topical probiotic treatments.

The findings present in these studies are significant for clinical practice. Probiotic-based topical formulations offer a natural, non-invasive approach to managing various skin conditions, including acne, eczema, and psoriasis. These products can be integrated into existing treatment regimens to enhance therapeutic outcomes and improve patient quality of life [19,22,26]. For example, a bioactive from *Levilactobacillus brevis* DSM17250 has been found to stimulate the growth of *S. epidermidis*, promoting a healthy skin microbiome [60]. Later, *L. reuteri* DSM 17938 was used to develop an ointment that has shown promise in reducing the severity of atopic dermatitis symptoms [43]. Similarly, synbiotic baths have been shown to improve atopic dermatitis symptoms, providing a novel approach to managing this chronic condition [45]. Furthermore, an international study involving 396 patients has validated the effectiveness of topical prebiotics and postbiotics in improving the symptoms of atopic dermatitis [61].

Other studies further support the efficacy of probiotic-based treatments. The application of *L. plantarum* LB244R ointment has been shown to alleviate skin aging, demonstrating significant improvements in skin elasticity and hydration [41]. Moreover, the Dermocosmetic Mineral 89 Probiotic Fractions adjunct to topical retinoids has shown anti-aging benefits, including improved skin hydration and reduced irritation [24].

The use of probiotic skin creams has also been found to improve outcomes following fractional CO_2_ laser resurfacing, suggesting that probiotics can enhance skin recovery and reduce post-procedure complications [62]. Additionally, the safety and tolerability of topical probiotic formulations containing *Micrococcus luteus* Q24 have been confirmed in healthy adults, indicating their potential for broader dermatological applications [63].

The selective targeting of skin pathobionts and inflammation with topically applied lactobacilli has demonstrated significant reductions in skin inflammation and pathogen load, highlighting the therapeutic potential of probiotics in managing skin microbiome imbalances [64]. The use of a lotion containing the heat-treated probiotic strain *L. johnsonii* NCC 533 has been effective in reducing *S. aureus* colonization in atopic dermatitis, underscoring the role of probiotics in managing bacterial skin infections [26]. Additionally, a topical anti-acne cream containing postbiotics has demonstrated efficacy in treating mild-to-moderate acne, providing an effective alternative to traditional treatments [65].

The characterization of live *C. acnes* subspecies *defendens* strain XYCM42 has shown promise as a topical regimen for promoting general skin health and cosmetic benefits, further expanding the potential applications of probiotics in skincare [11]. Additionally, natural topical treatments (based on probiotic extracts, honey, turmeric, and vitamin B12) have been shown to contribute to a reduction in dry scalp symptoms in children, showcasing the versatility of probiotics in addressing various dermatological conditions [66].

Topical creams containing live lactobacilli have the ability to reduce the population of malodor-producing bacteria and downregulate the expression of genes encoding PLP-dependent enzymes in the axillary skin microbiome, suggesting the potential of probiotics in managing body odor [9]. An emulsion formulated using *Lactococcus lactis* enhanced the skin barrier function in reconstructed human epidermis models, underscoring the potential of probiotics in improving overall skin health [67]. Similarly, a lotion containing probiotic ferment lysates (postbiotic product) has been found to bolster the skin barrier, thereby providing additional benefits for skin well-being [46].

Despite covering 106 studies on the use of topical probiotics, this review presents some limitations. Firstly, there is a lack of uniformity among the studies in terms of the specific dermatological conditions addressed, which complicates direct comparison of the results. Additionally, most studies focus on limited areas, such as the face and extremities, restricting the understanding of the impact of probiotics on less-studied regions of the skin, such as the scalp or intertriginous areas. Another limiting factor is the absence of a systematic approach that broadly considers factors such as age, sex, and geographical location, which can affect the efficacy of topical probiotics. Furthermore, this review was restricted to using a single database, the Web of Science, which may have resulted in the exclusion of relevant studies available on other indexing platforms, thereby limiting the comprehensiveness of the research. Many studies are of short duration, preventing a more robust analysis of the long-term effects of these treatments.

The analysis explores three key research domains involving topical probiotic applications: wound healing, dermatological conditions, and cosmetic skincare. However, these areas exhibit distinct methodological approaches, target populations, and stages of scientific progress. Specifically, research on wound healing therapies remains predominantly in the experimental phase, with limited translation to clinical practice observed thus far [32,44]. In contrast, studies on skin diseases generally adopt a more clinical approach, centering on specific dermatological interventions for inflammatory conditions [45,61,62]. Conversely, skin care research encompasses both in vitro and clinical investigations, though frequently limited by modest participant numbers [11,59,68]. Therefore, grouping the articles on three distinct topics related to topical probiotic use constitutes a limitation of the study.

The analyzed studies collectively highlight the substantial potential of probiotic-based topical formulations in the field of dermatology. These products offer a range of beneficial effects, including anti-aging and anti-inflammatory properties, as well as improvements in skin barrier function and microbiome balance. The findings presented support the integration of probiotic-based treatments into clinical practice, with the aim of enhancing therapeutic outcomes and improving the quality of life for patients.

## 5. Future Directions

This bibliometric study examined the topical use of probiotics for the treatment of inflammation and general skin care. The analysis evaluated factors such as co-authorship, citations, bibliographic coupling, and word co-occurrence. The findings indicate that research on topical probiotics has global relevance and impact, as evidenced by cross-country collaborations. The studies cover a diverse range of applications, from odor control to atopic dermatitis and environmental skin protection, highlighting the broad benefits of probiotics in dermatology. The collaborative nature of this field, evident from the analysis of citations from top-contributing countries, underscores the global importance and the need for continued joint efforts to drive innovation and practical applications of probiotics in skin care.

Despite the advancements in topical probiotic research, there are still important challenges that warrant careful consideration and further investigation. The selection of appropriate probiotic strains for specific skin conditions is a critical factor, as different microbial species and strains can have varying effects on the skin microbiome and host response. Future research should focus on developing and testing targeted topical probiotic formulations to address specific dermatological conditions such as acne, dermatitis, eczema, and psoriasis. Identifying and validating probiotic strains is crucial. For acne, strains like *L. reuteri* and *C. acnes* subspecies *defendens* should be investigated for their ability to reduce inflammation and regulate sebum production. For dermatitis and eczema, strains such as *L. rhamnosus* and *B. longum* should be prioritized to enhance skin barrier function and reduce inflammatory responses. In the case of psoriasis, the efficacy of *L. sakei* and *L. plantarum* in modulating immune responses and alleviating psoriatic symptoms should be explored.

Developing stable formulations that maintain the viability and efficacy of probiotics throughout the product’s shelf life is essential. This includes testing different delivery systems, such as creams, gels, serums, and lotions, to determine the most effective method for various skin conditions. Conducting well-designed clinical trials is necessary to establish the safety and efficacy of topical probiotics. Long-term studies are particularly important for understanding the sustained effects of these treatments on chronic conditions like acne and dermatitis. For example, a clinical trial could evaluate the long-term benefits of an *L. reuteri* cream in reducing acne lesions and preventing recurrence.

Investigating the mechanisms by which probiotics exert their beneficial effects on the skin is another critical area. This includes studying how probiotics interact with the skin microbiome, modulate the immune response, and enhance the skin barrier. For instance, research could focus on how *B. longum* strengthens the skin barrier by promoting ceramide production and reducing transepidermal water loss. Examining how factors such as age, sex, and geographical location influence the effectiveness of probiotic treatments can lead to more personalized skincare solutions, improving outcomes for diverse populations. Personalized treatments could involve tailoring probiotic formulations based on individual skin microbiome profiles and specific dermatological needs.

Expanding research to less-studied areas of the skin, such as the scalp and intertriginous regions, can provide a more comprehensive understanding of the benefits of topical probiotics. Studies could explore the use of probiotics in treating scalp conditions like seborrheic dermatitis or intertriginous dermatitis in skin folds. Exploring the potential of combining probiotics with other therapeutic agents, such as retinoids or anti-inflammatory compounds, can enhance treatment efficacy and provide synergistic benefits for managing complex skin conditions. For example, a combination therapy using *L. johnsonii* and topical corticosteroids could be tested for its effectiveness in reducing atopic dermatitis flare-ups.

The advancement of research on probiotics and skin health relies heavily on interdisciplinary collaboration. Partnerships between dermatologists, microbiologists, pharmacologists, and industry experts are crucial for driving innovation and translating research findings into effective clinical and commercial applications. Such collaborations are essential for fostering scientific progress and improving patient outcomes, as well as facilitating the development of standardized protocols and enhancing the reproducibility and reliability of research findings.

## 6. Conclusions

This bibliometric analysis examined the topical use of probiotics for wound healing, inflammation, and general skin care. This included evaluating co-authorship, citations, bibliographic coupling, and word co-occurrence. Research on topical probiotics has global relevance and impact, as shown by cross-country co-authorship analysis. Studies cover a range of applications, from odor control to atopic dermatitis and environmental skin protection, highlighting the diverse benefits of probiotics in dermatology. Collaboration across countries integrates knowledge and technologies, enabling innovation and effective skin health solutions. Analysis of citations from the top 10 countries reveals a diverse, collaborative landscape, reflecting the global importance of this field and the need for continued collaborative efforts to drive innovation and practical application of probiotics in skin care.

The co-authorship network indicated a collaborative research structure, with distinct groups of institutions from Greece, France, Brazil, and South Korea leading the collaboration. The strong presence of cosmetic companies underscores the industry’s importance in dermatological research. The studies explored a range of applications, from wound healing and reducing skin bacterial colonization to anti-aging benefits and treating allergic skin inflammation. Both in vivo, in vitro, and clinical investigations emphasized the value of combined probiotic formulations and their potential for therapeutic and cosmetic uses. Ranking analysis identified influential institutions, such as Shiraz University, Nestlé Research Center, and Malmö University, based on high average citations per article. International collaborations and university–industry partnerships are crucial for driving significant advances in topical probiotic research and fostering innovation in skin care applications.

The co-authorship network analysis revealed a diverse collaborative structure among the researchers, with multiple clusters engaged in various aspects of the research. A temporal assessment of the collaborations indicated active contributions through recent publications, highlighting the dynamic and expanding nature of this field of study. Intense collaborations within the green and red clusters suggest cohesive and productive research groups, while the blue and yellow clusters exhibit more specific and focused collaborations. These collaborative networks are essential for driving significant advancements in topical probiotic research, fostering innovation, and making new discoveries in the realm of skin care.

This study highlighted the importance of international and institutional collaboration, identifying key contributors and trends in research on using probiotics topically. The findings emphasize the relevance of ongoing collaborative efforts to promote innovations and improve skin health through probiotic use, leading to significant advancements in dermatology and enhanced patient outcomes.

## Figures and Tables

**Figure 1 microorganisms-12-02010-f001:**
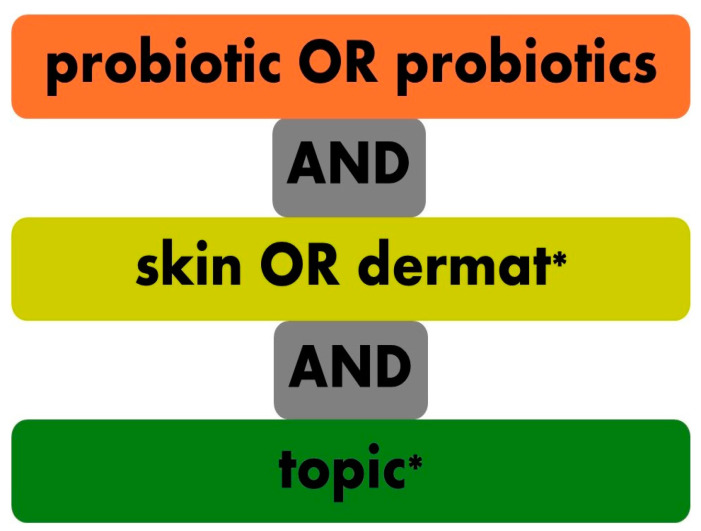
Keywords used in the advanced search of the Web of Science.

**Figure 2 microorganisms-12-02010-f002:**
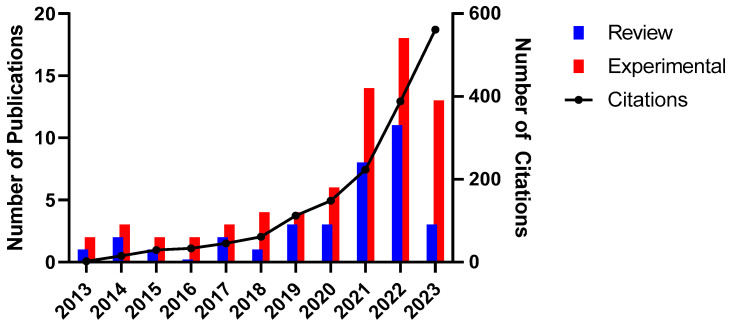
Total publications and citations per year.

**Figure 3 microorganisms-12-02010-f003:**
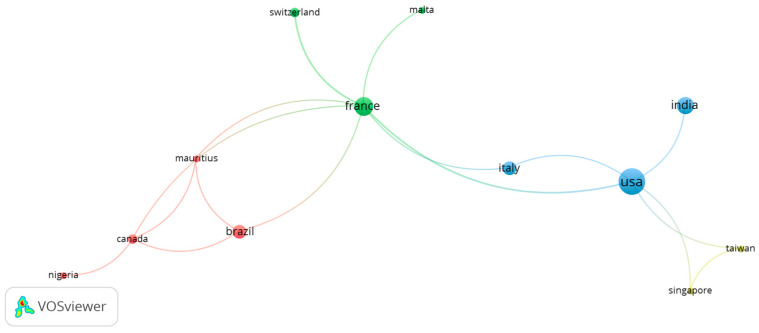
International collaboration networks among countries conducting research on the use of probiotics for wound healing, inflammation, and skin care. The countries actively participating in this research are represented by nodes, with their size indicating the number of publications from each country. The connecting lines between countries represent their collaborations in scientific publications, and the thickness of the lines reflects the intensity of these international partnerships. This color-coded visualization depicts collaboration clusters, representing groups of countries that frequently collaborate on scientific publications.

**Figure 4 microorganisms-12-02010-f004:**
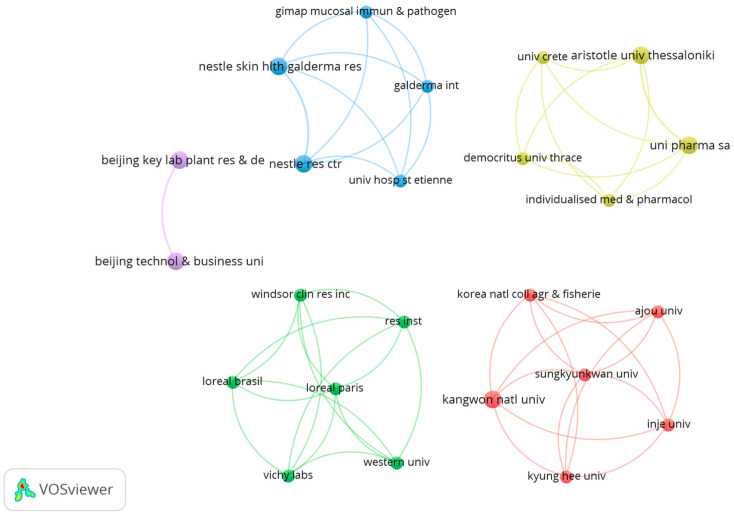
The visual depicts the 19 organizations at the forefront of experimental collaborative research on the topical use of probiotics. The node sizes correspond to publication volume, while the color groupings signify the primary collaboration clusters. This map elucidates the key network driving advancements in topical probiotic research, underscoring the significance of strategic partnerships in fostering scientific discoveries and new therapeutic possibilities. The different colors represent collaboration clusters, indicating groups of institutions that frequently collaborate on scientific publications. Institutions with a total link strength greater than or equal to 4 were selected.

**Figure 5 microorganisms-12-02010-f005:**
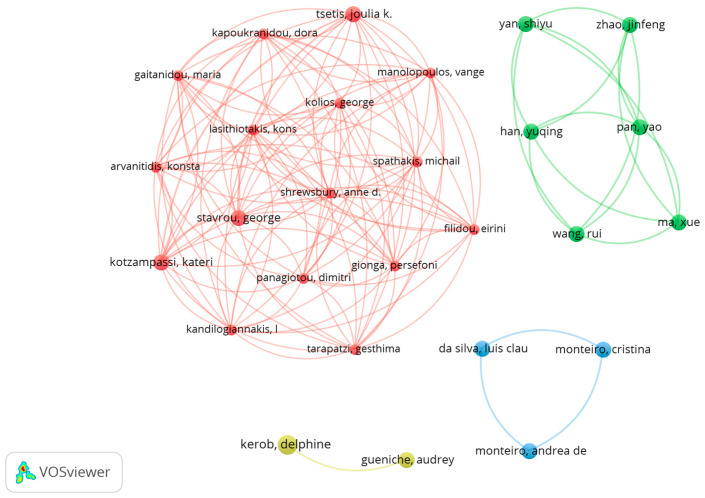
Co-authorship map between authors who have published articles on the topical use of probiotics in wound healing processes, skin inflammation, and general skin care. The colored clusters represent groups of authors who have collaborated closely. The node size corresponds to each author’s publication output, and the lines between nodes indicate collaborations between authors. Thicker lines indicate a stronger collaboration intensity. Authors with a total link strength greater than or equal to 15 were selected.

**Figure 6 microorganisms-12-02010-f006:**
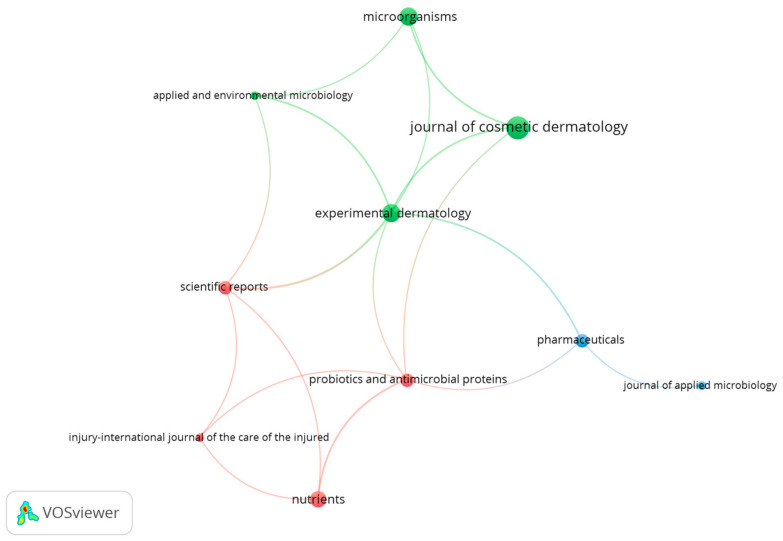
Citation mapping of the top 10 journals in research on dermatology with probiotics. This citation map visualizes the relationships among the 10 most influential journals in probiotic dermatology research. The graph shows how these journals are interconnected, citing each other, and contributing to the field. The distinct colored clusters represent groups of closely related journals based on their citation patterns. The size of the nodes reflects the number of publications in each journal, highlighting the collaborative nature of scientific research and knowledge dissemination through these influential publications.

**Figure 7 microorganisms-12-02010-f007:**
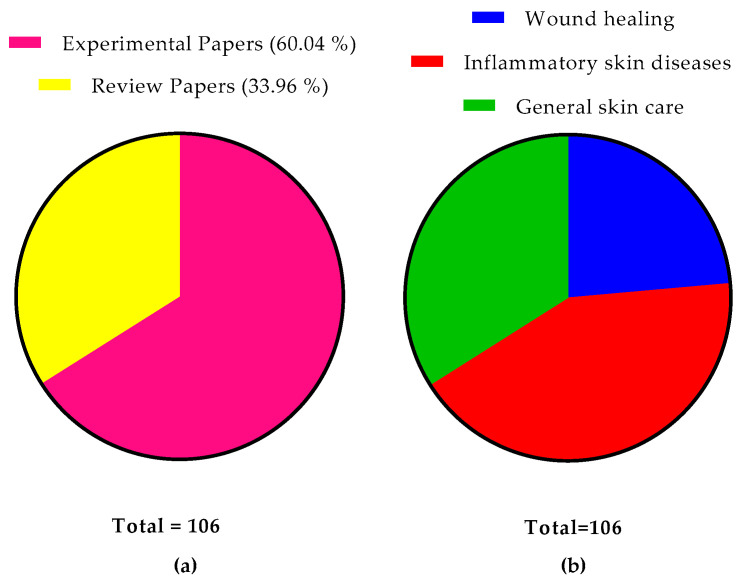
(**a**) Categorization into experimental and review papers. (**b**) Categorization by application proposals.

**Figure 8 microorganisms-12-02010-f008:**
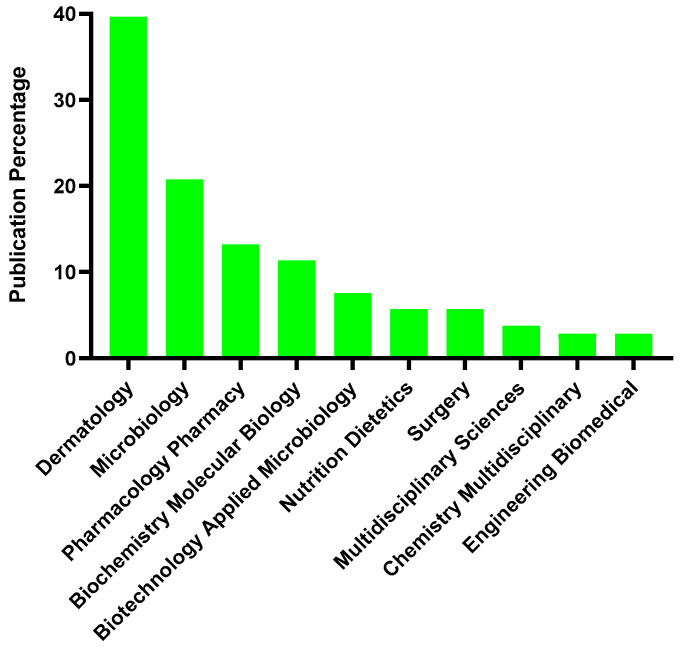
Top 10 Web of Science Categories.

**Figure 9 microorganisms-12-02010-f009:**
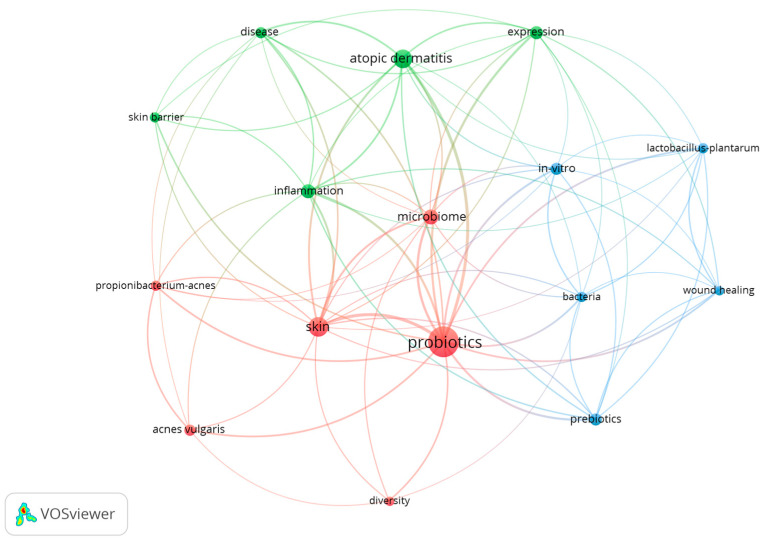
Keyword co-occurrence network in experimental articles on the topical use of probiotics on the skin. The keywords presented are those that appear at least five times within the data. The visual nodes depict these keywords, with their size proportional to the frequency of occurrence. The cluster colors represent distinct research domains. The connections between the nodes illustrate the co-occurrence of keywords across the analyzed articles, thereby elucidating the inter-relationships among the various research themes.

**Figure 10 microorganisms-12-02010-f010:**
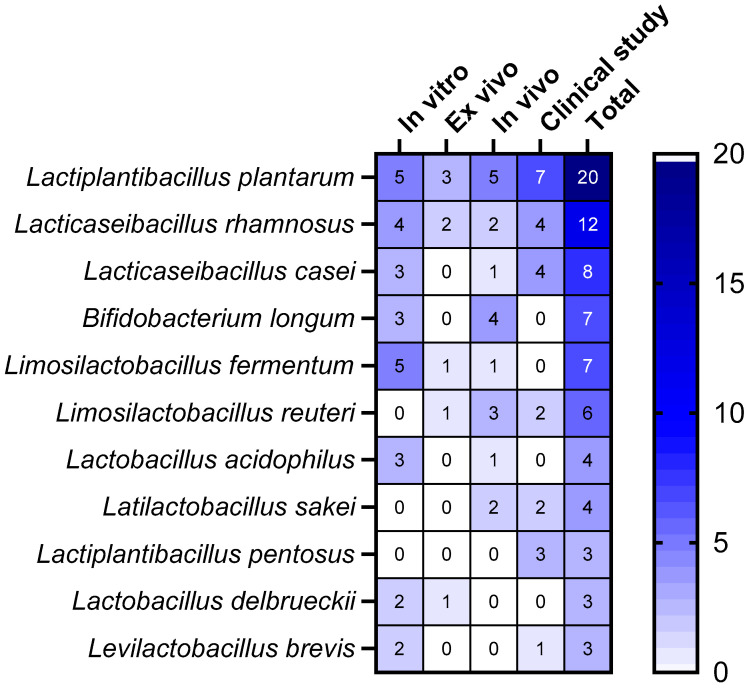
Probiotic strains most frequently used in topical skin care products according to the experimental studies reviewed.

**Figure 11 microorganisms-12-02010-f011:**
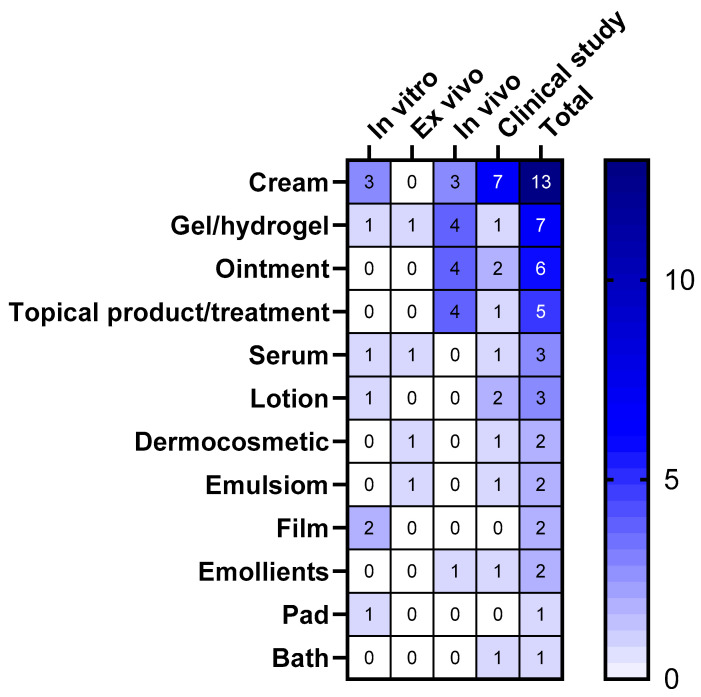
Probiotic-based topical formulations developed in the enrolled studies.

**Figure 12 microorganisms-12-02010-f012:**
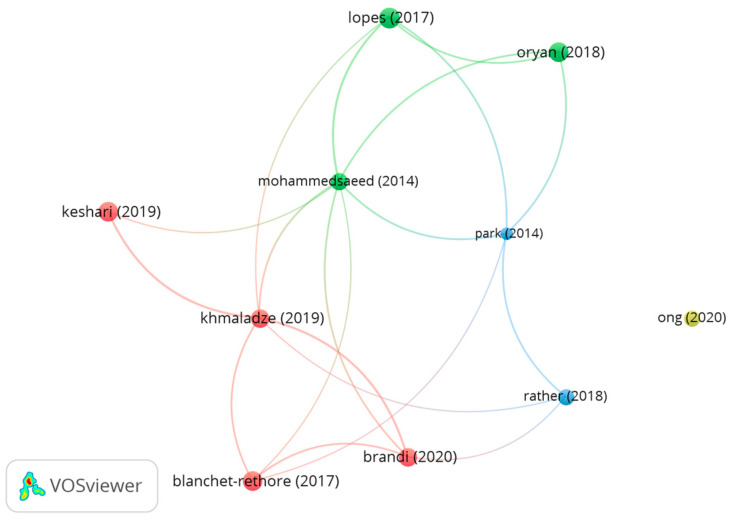
Bibliographic coupling network of experimental articles on the topical use of probiotics on the skin. The visualization depicts individual research articles as nodes, with node size indicative of the citation count for each article. The distinct colors of the nodes denote different clusters, which group together articles that share a high number of citations, implying a close thematic connection. Furthermore, the lines connecting the nodes represent the strength of bibliographic coupling, which illustrates the intensity of citations shared among the articles.

**Figure 13 microorganisms-12-02010-f013:**
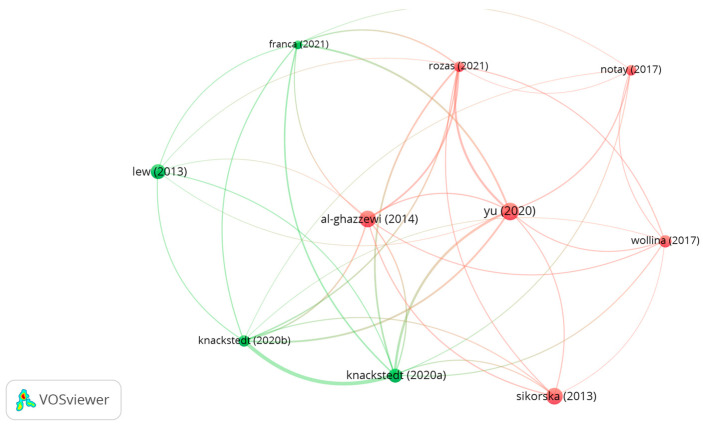
Bibliographic coupling network between the 10 most cited review articles on the use of topical probiotics in skin health. The nodes symbolize individual articles, with their size indicating the number of citations that each article has received. The cluster colors represent groupings of articles based on their bibliographic relationships. The lines connecting the nodes show the strength of bibliographic coupling between articles, reflecting the number of shared references.

**Table 1 microorganisms-12-02010-t001:** Top 10 countries by total citations in research on the topical application of probiotics. The ranking considers the number of experimental articles, the average citations per article, and the degree of international collaboration measured by strength total link (STL). The United States and France lead the list, followed by South Korea, Sweden, and Iran, all of which demonstrate high average citations per article, suggesting the significance and impact of their scholarly contributions in this domain.

Ranking	Countries	Papers	Citations	Citations per Paper	STL
1st	United States	13	108	8.31	6
2nd	France	7	94	13.43	9
3rd	South Korea	5	79	15.8	0
4th	Sweden	3	71	23.67	0
5th	Iran	2	67	33.5	0
6th	Switzerland	2	66	33	2
7th	Portugal	3	61	20.3	0
8th	Italy	4	60	15	2
9th	Singapore and Taiwan	1	49	49	2
10th	Japan	2	48	24	3

**Table 2 microorganisms-12-02010-t002:** The top 10 institutions or groups of institutions that are most frequently cited in experimental research on the topical use of probiotics. This table presents the number of articles published by each institution, the total citations received, the average citations per article, and the total link strength (STL) in citation mapping.

Organizations	Papers	Citations	Citations per Paper	STL
Shiraz University	2	67	33.5	0
Nestlé Research Center and NestléSkin Health Galderma R&D	2	66	33	5
Malmö University and BioGaia AB	2	63	31.5	3
Portuguese Catholic University	3	61	20.33	0
Yeungnam University	2	59	29.5	4
University Hospital of Saint-Etienne and Galderma Int and Mucosal Immune & Pathogen Agents Grp	1	50	50	4
National Central University and National University of Singapore and University of California San Diego	1	49	49	2
Manchester University	2	44	22	0
University of Eastern Piedmont and Verona University and Oriflame Cosmetics and Sintal Dietetics S.R.L.	1	43	43	2
Universiti Sains Malaysia and Inha University and Dongguk University and Perdana University and Ministry of Science, ICT and Future Planning and RIKEN	1	36	36	3

**Table 3 microorganisms-12-02010-t003:** Top 10 corresponding authors according to number of citations. The table shows the number of articles published by each author, the total citations received, the total citations excluding self-citations, the average number of citations per article, and the total link strength (STL) in the citation mapping.

Correspondents Authors	Papers	Citations		Citations per Paper	STL
		Total	Excluding Self-Citations		
A. Mercenier	2	66	65	33	12
E. Butler	2	63	62	31.5	5
F. K. Tavaria	3	61	60	20.33	8
Y.-H. Park	2	59	51	29.5	13
A. Oryan	1	51	50	51	3
S. Blanchet-Rethore	1	50	65	50	5
C.-M. Huang	1	49	41	49	5
C. A. O’Neill	2	44	41	22	5
D. Cecconi	1	43	41	14	43
J. M. Delgado-Lopez	2	25	22	11	12.5

**Table 4 microorganisms-12-02010-t004:** The 10 most cited journals in the area of topical application of probiotics. The table presents journals ranked by the number of citations that their experimental articles have received, along with the number of articles published, the average number of citations per article, and the total link strength (STL) in citation mapping.

Journals	Papers	Citations	Citations per Paper	STL
Experimental Dermatology	4	67	16.75	16
Journal of Applied Microbiology	1	55	55	4
Burns	1	51	51	1
Clinical Cosmetic and Investigational Dermatology	4	50	12.5	3
International Journal of Molecular Sciences	1	49	49	1
Probiotics and Antimicrobial Proteins	2	43	21.5	10
Scientific Reports	2	43	21.5	7
Applied and Environmental Microbiology	1	41	41	7
Microorganisms	4	39	9.75	6
Frontiers in Microbiology	1	36	36	1

**Table 5 microorganisms-12-02010-t005:** Top 10 most cited experimental papers.

Years	Authors	Title	Journal	Citations	
				Total	Excluding Self-Citations
2017	E.G. Lopes, et al.	Topical application of probiotics in skin: adhesion, antimicrobial and antibiofilm in vitro assays [51]	Journal of Applied Microbiology	55	54
2018	A. Oryan, M. Jalili et al.	The concurrent use of probiotic microorganism and collagen hydrogel/scaffold enhances burn wound healing: an in vivo evaluation [14]	Burns	51	50
2017	S. Blanchet-Réthoré et al.	Effect of a lotion containing the heat-treated probiotic strain *Lactobacillus johnsonii* NCC 533 on *Staphylococcus aureus* colonization in atopic dermatitis [26]	Clinical, Cosmetic and Investigational Dermatology	50	49
2019	S. Keshari et al.	Butyric Acid from Probiotic *Staphylococcus epidermidis* in the Skin Microbiome Down-Regulates the Ultraviolet-Induced Pro-Inflammatory IL-6 Cytokine via Short-Chain Fatty Acid Receptor [30]	International Journal of Molecular Sciences	49	43
2019	I. Khmaladze et al.	*Lactobacillus reuteri* DSM 17938—A comparative study on the effect of probiotics and lysates on human skin [22]	Experimental Dermatology	43	41
2020	J. Brandi et al.	Exploring the wound healing, anti-inflammatory, anti-pathogenic and proteomic effects of lactic acid bacteria on keratinocytes [18]	Scientific Reports	43	38
2014	W. Mohammedsaeed et al.	*Lactobacillus rhamnosus* GG Inhibits the Toxic Effects of *Staphylococcus aureus* on Epidermal Keratinocytes [39]	Applied and Environmental Microbiology	41	35
2019	J. S. Ong et al.	*Lactobacillus plantarum* USM8613 Aids in Wound Healing and Suppresses *Staphylococcus aureus* Infection at Wound Sites [48]	Probiotics and Antimicrobial Proteins	36	36
2018	I. A. Rather et al.	Probiotic *Lactobacillus sakei* proBio-65 Extract Ameliorates the Severity of Imiquimod Induced Psoriasis-Like Skin Inflammation in a Mouse Model [12]	Frontiers in Microbiology	36	30
2014	S. B. Park, M. I. et al.	Effect of Emollients Containing Vegetable-Derived *Lactobacillus* in the Treatment of Atopic Dermatitis Symptoms: Split-Body Clinical Trial [52]	Annals of Dermatology	23	19

**Table 6 microorganisms-12-02010-t006:** The top 10 most cited review articles.

Years	Authors	Titled	Journal	Citations	
				Total	Excluding Self-Citations
2020	Y. Yu et al.	Changing our microbiome: probiotics in dermatology [53]	British Journal of Dermatology	91	90
2014	F.H. Al-Ghazzewi and R.F. Tester	Impact of prebiotics and probiotics on skin health [19]	Beneficial Microbes	86	83
2013	H. Sikorska and W. Smoragiewicz	Role of probiotics in the prevention and treatment of methicillin-resistant *Staphylococcus aureus* infections [54]	International Journal of Antimicrobial Agents	84	84
2013	L.C. Lew and M.T. Liong	Bioactives from probiotics for dermal health: functions and benefits [55]	Journal of Applied Microbiology	65	62
2020	R. Knackstedt et al.	The role of topical probiotics in skin conditions: A systematic review of animal and human studies and implications for future therapies [17]	Experimental Dermatology	58	58
2016	U. Wollina	Microbiome in atopic dermatitis [56]	Clinical, Cosmetic and Investigational Dermatology	43	43
2020	R. Knackstedt et al.	The role of topical probiotics on wound healing: A review of animal and human studies [18]	International Wound Journal	38	38
2021	M. Rozas et al.	From Dysbiosis to Healthy Skin: Major Contributions of *Cutibacterium acnes* to Skin Homeostasis [10]	Microorganisms	37	35
2017	M. Notay et al.	Probiotics, Prebiotics, and Synbiotics for the Treatment and Prevention of Adult Dermatological Diseases [57]	American Journal of Clinical Dermatology	35	32
2021	K. França	Topical Probiotics in Dermatological Therapy and Skincare: A Concise Review [58]	Dermatology and Therapy	24	24

## Data Availability

The original contributions presented in the study are included in the article, further inquiries can be directed to the corresponding author.

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
