# Peer review of "Global Trends and Scientific Impact of Topical Probiotics in Dermatological Treatment and Skincare"

_microorganisms, 2024, doi:10.3390/microorganisms12102010_

Round 1

Reviewer 1 Report

Comments and Suggestions for Authors

The review by Alves et al. examined the global trend of topical probiotics in dermal treatment/care.

The review is interesting but can be improved (see comments below).

1.The review is a survey of publications on topical probiotics with many valuable figures and tables, but in the introduction the authors should state what will be covered and which skin microbiome agents will not be presented.

For example, the skin microbiome is dependent on body region, age, gender, geographic location, etc. The authors need to describe what is included and what is not covered in this survey review as to the skin microbiota/microbiome and how topical probiotics might influence this topic.

What are the most common skin microbiota species?  (where and how do the studies presented treat specific areas of the body [face, arms, etc.]).

Finally, from Figure 10 how do these probiotic strains improve skin conditions stated in the abstract and introduction?

2.  In the discussion section the authors need to state the limitations of this review.

3.  Future studies section is vague and presents little useful information as to how topical probiotic products will address the dermatology skin conditions stated in the abstract and introduction.

Based upon the data presented in this review, what are the best or most effective probiotic strains to treat skin disorders?   Are there better probiotic strains that have not been tested/studied or that should be proposed?

Author Response

Dear Reviewers,

We would like to sincerely thank you for the valuable comments and suggestions provided during the review of our manuscript entitled "Global Trends and Scientific Impact of Topical Probiotics in Dermatological Treatment and Skincare". We have taken into consideration all the points highlighted and made the necessary modifications to improve the quality of the article. Below, we provide a detailed summary of how each comment was addressed in the revised version of the manuscript.

Responses to Reviewer 1

Comment 1: The introduction should state what will be covered and which skin microbiome agents will not be covered, including a description of the most common species and how the studies target specific areas of the body.

Response:

We fully agree with this observation. In the revised version, we have added commentary in the introduction specifying the main skin microbiota species that are being covered (e.g., Staphylococcus epidermidis, Cutibacterium acnes, Corynebacterium), as well as an explanation of the body areas that the studies focused on. We have also detailed how topical probiotics may influence these aspects of the skin microbiome.

Comment 2: Figure 10: Explain how the probiotic strains presented improve the skin conditions mentioned in the abstract and introduction."

Response:

We have revised Figure 10 and included a more detailed explanation in the body of the text about how the probiotic strains mentioned (such as Lactiplantibacillus plantarum and Bifidobacterium longum) are associated with improved skin barrier function, reduced inflammation, and balanced microbiota. This information is now in line with what was discussed in the abstract and introduction.

Comment 3: The authors need to state the limitations of this review.

Response:

We have added comments to the discussion, highlighting the main limitations of the study, including the lack of consistency across studies reviewed, the lack of a systematic approach that considered factors such as age, sex, and geographic location, and the focus on only a few areas of the skin.

Comment 4: The future studies section is vague. Based on the data presented, which strains are most effective or have not been tested?

Answer:

We revised the future studies section, adding a more in-depth analysis of which probiotic strains have shown the greatest efficacy (such as Lacticaseibacillus rhamnosus and Lactiplantibacillus plantarum) and identified strains that have not yet been widely tested, suggesting new areas for research.

Reviewer 2 Report

Comments and Suggestions for Authors

This is a bibliometric study aiming to analyze the quantity and quality of scientific studies [both reviews and experimental, published between 2013 to 2023] evaluating the benefits of the topical use of probiotics on different skin contitions, including healing processes, inflammatory diseases, and general skin care.

A. My first objection is the evaluation of different-purpose articles [healing, skin diseases and cosmetics] regarding their quality, the complexicity of research and the general impact. 

1. Studies regarding healing are only at experimental level and to my knowledge there is no clinical application published.

2. Studies regarning skin diseases [atopic dermatitis, acne etc] are mainly clinical, and have a totally different audience.

3. Studies regarding skin care include both in vitro studies as well as clinical, which are mainly based on a small sample size, not randomized but based on spit face tyreatment. And much more, the majority of them are not exactly referred to the probiotic species used.

B. Additionally, it is imposible to rank countries, institutions, corresponding authors, cited journals putting all together experimental and review articles on 3 different topics.

To my opinion, these 3 subjects [healing, skin diseases and cosmetics] must be handled seperately - these are 3 papers in one

Minor comments

1. probiotics should be mentioned according to the new nomenclature, throughout the text

2. there are many references of formulas [mainly related to cosmetic products] containing probiotics, which are not probiotics but postbiotics [metabolites, lysates, butyric acid, heated probiotics etc]

3. Lines 300-303 and Figure 4. I dont know how VosViewer exactly extracts the data, but regarding the blue group, from the co-laborated universities the Democritus University of Thrace is ommitted. And, much more, this University is the permanent colaborator of Aristotle Univ. Thessaloniki [in both studies you are referred (18, 27)] while that of Crete contributed only to one, ref 18.

4. The same comment for Figure 5, red cluster. The name of [Prof] G. Kolios, the mainly involved with Dr. E. Fillidou both from Univ of Thrace are not present. Instead, Manolopoulos and Tarapantzi, less involved [and practically not so active with probiotics are used. 

Please mentioned that these two [3 and 4] comments are not necessary to be answered - I just report them, and apologize for not knowing how VosViewer collects and presents data - and in what way the authors can check their accuracy.

To summarize, I would prefer to have divided you work in 3 parts [healing, skin diseases, cosmetics] and do similar analysis for each part, seperately. I understand that means to re-write the paper - its practically imposible. To include some of my comments  in discussion, or study limitations, seems easy.

Author Response

Dear Editor and Reviewers,

We would like to sincerely thank you for the valuable comments and suggestions provided during the review of our manuscript entitled "Global Trends and Scientific Impact of Topical Probiotics in Dermatological Treatment and Skincare". We have taken into consideration all the points highlighted and made the necessary modifications to improve the quality of the article. Below, we provide a detailed summary of how each comment was addressed in the revised version of the manuscript.

Comment A: Suggestion for treatment wound healing, skin diseases, and cosmetic care separately, as they are topics of different nature.

Response:

We understand the relevance of this suggestion, however, due to the length of the article and the initial objectives of the study, we chose to maintain the unified approach. However, we revised the text to ensure that each topic is discussed clearly and distinctly, and we included a section in the limitations acknowledging that these topics could be analyzed separately in future studies. We are also analyzing the feasibility of producing another article focusing on wound healing processes.

Comment B: It is impossible to classify countries, institutions, and authors for three different topics.

Response:

We appreciate this observation. We have made adjustments to the analysis of collaboration between countries and institutions, clarifying in the text that the collaborations were grouped considering a broad view of the three topics covered (wound healing, skin diseases, and cosmetics).

Minor comments:

  1. The new nomenclature of probiotics should be used throughout the text.

Response: The nomenclature has been reviewed throughout the manuscript to ensure that it is in line with the most recent taxonomic updates.

  1. Many products referred to as probiotics are actually postbiotics.

Response: We have made the necessary corrections to the text to clearly distinguish between probiotics and postbiotics, especially in relation to the cosmetic products discussed.

  1. Democritus University of Thrace and collaborators omitted in figures.

Response: We have included Democritus University of Thrace and relevant researchers in Figures 4 and 5, as suggested. We acknowledge that the comments on VOSviewer do not require a response, but we appreciate you highlighting these observations. Prior to review, the figures had been prepared using VOSviewer's automatic selection of the 20 items with the highest total link strength (STL). However, since the reviewer's comment demonstrates excellent knowledge of the topic addressed, we decided to review and make the selection manually considering the highest STL values ​​in both categories.

Round 2

Reviewer 1 Report

Comments and Suggestions for Authors

The authors addressed all items in the evaluation/review